REGISTERED REPORT PROTOCOL

# Effect of stem cell therapies on tendon-bone healing after anterior cruciate ligament reconstruction in animal models: Protocol for a systematic review and meta-analysis

**Shibo Zhao, Congcong Wang** [ID] *

Department of Joint Surgery, Weifang People's Hospital, Shandong Second Medical University Weifang, Shandong, China

* 1413138161@qq.com

This is a Registered Report and may have an associated publication; please check the article page on the journal site for any related articles.

---

## Abstract

### Introduction

Anterior cruciate ligament (ACL) reconstruction is widely performed, yet insufficient tendon-bone healing remains a key contributor to graft failure. Stem cell-based interventions, including mesenchymal stem cells (MSCs) and stem cell-derived products (e.g., exosomes/extracellular vesicles), have shown potential to enhance tendon-bone integration in preclinical models. However, findings across animal studies are heterogeneous and have not been comprehensively synthesized. This review aims to evaluate the effects of stem cell-based therapies on tendon-bone healing after ACL reconstruction in animal models.

### Methods and analysis

This protocol follows PRISMA-P. We will search PubMed, Embase, Scopus, SPORT-Discus, Web of Science, and the Cochrane Library from inception to the final search date. Controlled animal studies comparing ACL reconstruction with stem cell-based interventions versus controls will be included. Primary outcomes are biomechanical properties (ultimate failure load and stiffness). Secondary outcomes include micro-CT measures of bone integration (e.g., bone volume fraction, bone mineral density) and histological outcomes (e.g., interface maturation or validated scoring systems). Risk of bias will be assessed using SYRCLE's tool. Random-effects meta-analyses will be performed, with prespecified subgroup analyses by stem cell type, delivery method, animal species, and follow-up time. Sensitivity analyses and publication-bias assessments will be conducted where appropriate.

### PROSPERO registration

CRD420251137985.

**Data availability statement:** The search strategy used for this systematic review is provided in S2 Table. The data underlying the results (e.g., the final extracted dataset, risk-of-bias table, and analysis code) will be made publicly available upon publication of the final meta-analysis article. The data will be shared via a public data repository such as Open Science Framework (OSF) or Zenodo and will be accessible in the Supporting information files or through a direct repository link at that time.

**Funding:** Shandong Province medical health science and technology project (202404070285), Research on the Mechanism and Application of Microfragment Adipose tissue in promoting Tendon-bone Healing after anterior cruciate ligament Reconstruction The funder provided financial support for the study design and were involved in the original drafting of the manuscript, but they did not participate in data collection and analysis, decision to publish, or manuscript preparation.

**Competing interests:** The authors have declared that no competing interests exist.

## 1. Introduction

Anterior cruciate ligament (ACL) injury is one of the most prevalent musculoskeletal conditions, particularly among young and physically active individuals [1]. It is estimated that more than two million ACL injuries occur worldwide each year, and the incidence continues to rise in line with increased participation in high-impact sports. The standard treatment for complete ACL rupture is surgical reconstruction, which aims to restore joint stability, allow patients to return to pre-injury activity levels, and reduce the risk of secondary osteoarthritis. Despite advances in surgical techniques and postoperative rehabilitation protocols, failure rates after ACL reconstruction remain between 10% and 20%, with poor tendon-bone healing being one of the primary contributors to graft insufficiency [2–4]. Inadequate tendon-bone integration compromises graft stability, leads to bone tunnel enlargement, and predisposes patients to re-injury or chronic joint instability, which ultimately impairs long-term functional outcomes.

To address these challenges, stem cell-based therapies have emerged as one of the most promising approaches in regenerative medicine. Mesenchymal stem cells (MSCs), derived from bone marrow, adipose tissue, synovium, tendon, or umbilical cord blood, have shown potential in enhancing tendon-bone healing by promoting bone formation, modulating inflammation, and accelerating revascularization. In addition to direct cell transplantation, stem cell-derived products such as exosomes and extracellular vesicles have also been investigated, showing encouraging results in preclinical settings [5–8].

A growing body of animal studies has evaluated the efficacy of stem cell therapies for tendon-bone healing in ACL reconstruction models [5]. Some studies have reported improved biomechanical strength and greater bone tunnel mineralization following MSC administration, while others have demonstrated enhanced collagen fiber alignment and histological maturation after adipose- or tendon-derived stem cell delivery. However, findings remain inconsistent due to variations in stem cell sources, delivery methods, dosages, and follow-up durations.

Despite these limitations, the preclinical evidence suggests that stem cell therapies hold promise for improving tendon-bone healing after ACL reconstruction. However, no systematic review or meta-analysis has yet synthesized animal data to provide a comprehensive evaluation. This study aims to fill that gap by conducting a systematic review and meta-analysis of animal studies, focusing on biomechanical, histological, and imaging outcomes. By pooling these outcomes across various models, we aim to assess the efficacy of stem cell therapies and explore potential moderators such as stem cell type, delivery strategy, animal species, and follow-up duration.

## 2. Methods and analysis

### 2.1. Protocol registration and reporting standards

This systematic review will be conducted and reported in accordance with the Preferred Reporting Items for Systematic Reviews and Meta-Analyses (PRISMA) guidelines [9,10]. The present protocol follows the PRISMA-P 2015 statement (S1 Table).

The review has been registered in the PROSPERO (CRD420251137985). The search will be conducted from inception to the final search date, which will be updated immediately prior to manuscript submission. Any amendments to this protocol will be documented and reported in the final review.

## 2.2. Eligibility criteria

Inclusion Criteria: We will include controlled animal studies that evaluate the effect of stem cells or stem cell-derived products on tendon-bone healing after anterior cruciate ligament (ACL) reconstruction. Eligible interventions will include mesenchymal stem cells (MSCs) from bone marrow, adipose tissue, tendon, synovium, or umbilical cord blood, as well as their derivatives such as exosomes, extracellular vesicles, or scaffold-based delivery systems. However, it is important to note that scaffold-based delivery systems are only included when used in combination with stem cells or their derivatives, such as exosomes or extracellular vesicles. This clarification ensures that the study focuses specifically on stem cell-assisted interventions, rather than standalone scaffold therapies. Comparators must be ACL reconstruction without stem cell intervention, sham operation, or placebo/vehicle control. Studies will be considered if they report at least one relevant outcome measure, including biomechanical testing (e.g., ultimate failure load, stiffness), micro-CT evaluation of bone integration, or histological assessment of tendon-bone healing, and provide quantitative data. Outcomes will be categorized a priori as primary or secondary. The primary outcomes are biomechanical properties reflecting tendon-bone healing, including ultimate failure load and stiffness. Secondary outcomes include micro-CT-based measures of bone integration (e.g., bone volume fraction and bone mineral density) and histological outcomes (e.g., interface maturation and histological scoring results).

Exclusion Criteria:We will exclude clinical trials, human case reports, reviews, editorials, and letters. In vitro studies without animal models, studies without a control group, and those not reporting tendon-bone healing outcomes will also be excluded. Furthermore, duplicate publications, conference abstracts with insufficient data, and studies for which the full text cannot be retrieved will not be considered. We will exclude studies for which the full text cannot be retrieved. However, attempts will be made to obtain the full text through author contact or interlibrary loan before exclusion.

## 2.3. Search strategies

A systematic search will be conducted in PubMed, Embase, Scopus, SPORTDiscus, the Cochrane Library, and Web of Science from inception to the date of the final search. Only English-language articles will be included in the search. The following key terms and subject headings will be used in combination with Boolean operators: ("anterior cruciate ligament" OR ACL) AND (reconstruction OR surgery OR graft) AND ("stem cell" OR MSC OR "mesenchymal stem cell" OR "adipose-derived stem cell" OR "bone marrow stem cell" OR exosome OR extracellular vesicle OR scaffold) AND (animal OR rat OR rabbit OR preclinical OR in vivo). Additionally, we will systematically use controlled vocabulary (e.g., MeSH terms in PubMed, Emtree terms in Embase) to ensure comprehensive coverage. Reference lists of the included articles will also be screened manually to identify additional eligible studies.The complete search strategies for each database are provided in S2 Table.

## 2.4. Selection of studies

EndNote V.21 citation manager will be used to import all retrieved references and to remove duplicates. Two reviewers (SZ and CW) will independently screen the titles and abstracts of all publications identified through the electronic searches to determine eligibility according to the predefined inclusion criteria. Full texts of potentially relevant studies will then be retrieved and assessed for eligibility.

Reasons for exclusion at the full-text stage will be recorded in detail. Any disagreements between the two reviewers will be resolved through discussion, and if necessary, a third reviewer will be consulted. Studies that meet the eligibility criteria will be retained for data extraction. The study selection process will be documented in sufficient detail to enable the construction of a PRISMA 2020 flow diagram.

## 2.5. Data extraction

Data extraction for the eligible full-text articles will be conducted independently by two reviewers (SZ and CW). A pre-specified data extraction form will be used to ensure consistency and minimise discrepancies. Any disagreements will be resolved through discussion, and if necessary, the corresponding author of the study will be contacted for clarification. The following information will be collected: first author, year of publication, country, animal species, sample size, follow-up duration, type and source of stem cells or stem cell-derived products, dose/amount administered (e.g., total number of cells delivered; and, where applicable, concentration/volume and number of administrations to enable derivation of total dose), delivery method (injection, scaffold, graft wrapping), control intervention, outcome measures (biomechanical strength, micro-CT evaluation, histological assessment), and risk of bias assessment.

## 2.6. Quality assessment

Two reviewers (SZ and CW) will independently assess the methodological quality and risk of bias of the included animal studies using the SYRCLE's risk of bias tool, which is specifically designed for animal experiments [11]. The tool evaluates ten domains, including sequence generation, baseline characteristics, allocation concealment, blinding, random housing, random outcome assessment, incomplete outcome data, selective outcome reporting, and other sources of bias. Each domain will be judged as 'low risk,' 'high risk,' or 'unclear risk.' Any discrepancies will be resolved by discussion, with arbitration by a third reviewer if necessary. No studies will be excluded solely based on the quality assessment, but the risk of bias results will be considered in the interpretation of findings. Inter-reviewer agreement for the risk of bias assessment will be evaluated using Cohen's kappa statistic prior to consensus discussions, and the level of agreement will be reported.

## 2.7. Statistical analysis and data synthesis

The extracted data will first be entered into Microsoft Excel and then exported to STATA software (V.18) and Review Manager (RevMan V.5.4) for analysis. The characteristics of the included studies will be summarised in tables and descriptive text. Meta-analysis will be conducted using a random-effects model to estimate pooled effect sizes with 95% confidence intervals (CIs). Outcomes will be synthesized according to a prespecified hierarchy. Meta-analyses will prioritize the primary outcomes (ultimate failure load and stiffness). Secondary outcomes will be synthesized when at least two studies report sufficiently comparable data, including micro-CT measures (e.g., bone volume fraction and bone mineral density) and histological outcomes (e.g., histological scoring results). Effect sizes will be calculated as standardized mean differences (SMDs) when outcomes are measured using different scales; where outcomes share the same units across studies, mean differences (MDs) may be used to improve interpretability. Results will be presented as forest plots.

We anticipate heterogeneous interventions (different stem cell sources/products and delivery strategies). Therefore, meta-analyses will be conducted separately by intervention category, primarily: (1) live stem cells (e.g., MSCs from different tissues) and (2) stem cell-derived products (e.g., exosomes/extracellular vesicles). A pooled analysis of "any stem cell-based therapy" will be presented only when studies are judged sufficiently comparable in terms of intervention category, control condition, outcome definition, and follow-up timing; otherwise, findings will be summarized narratively. For multi-arm studies with a shared control group, we will avoid double-counting by combining eligible intervention arms into a single comparison when appropriate (e.g., different doses within the same category). If arms must be kept separate (e.g., live cells vs exosomes), we will split the control group evenly across comparisons.

For histological outcomes, we anticipate that different studies may use diverse semi-quantitative scoring systems to evaluate tendon-bone interface healing. Therefore, histological outcomes will be quantitatively synthesized only when they assess comparable constructs (e.g., interface maturation, fibrocartilage formation, collagen organization, or overall histological healing). When different scoring systems are used to assess similar constructs, standardized mean differences

(SMDs, Hedges' g) will be applied to enable pooling. If multiple histological domains are reported within the same study, we will extract and synthesize the prespecified overall histological score where available; otherwise, the domain most consistently reported across studies will be prioritized to minimize selective inclusion. When histological definitions or scoring systems are judged to be too heterogeneous for meaningful quantitative synthesis, results will be summarized narratively.

Dose-response (meta-regression). If sufficient studies are available (typically ≥10), we will conduct random-effects meta-regression to explore whether cell dose (total number of cells administered; log-transformed if appropriate) explains between-study heterogeneity in the primary outcomes. If dose is reported in incomparable units across studies, meta-regression will be restricted to studies reporting a derivable total cell number; otherwise, dose effects will be summarized narratively.

Subgroup analyses will be conducted a priori by therapy format (whole-cell therapies vs stem cell-derived products, e.g., exosomes/extracellular vesicles), and by cell source for whole-cell therapies (bone marrow-derived, adipose-derived, tendon-derived, synovial-derived, and umbilical cord/umbilical cord blood-derived cells). Additional subgroup analyses will be performed by delivery method (injection, scaffold-based delivery, graft wrapping), animal species (rat vs rabbit), and follow-up duration (≤4 weeks, >4 to ≤8 weeks, >8 to ≤12 weeks, and >12 weeks.), subject to data availability. In studies reporting multiple outcome time points, we will prioritize the longest follow-up period. If multiple time points are reported, separate analyses will be conducted for each relevant time point to evaluate temporal effects.

Statistical heterogeneity will be assessed using the $\chi^2$ test and quantified with the $I^2$ statistic. If substantial heterogeneity is detected ($I^2 > 75\%$) and sufficient data are available, potential sources of heterogeneity will be explored through subgroup analyses and sensitivity analyses. If substantial heterogeneity persists despite these efforts, we will consider reporting the results narratively rather than conducting a quantitative synthesis.

To address translational limitations related to the use of animal models, we will conduct prespecified sensitivity analyses stratified by animal species and experimental model characteristics. Specifically, analyses will be repeated after excluding individual species (e.g., rat-only or rabbit-only analyses) to evaluate the robustness of the pooled estimates. Where sufficient data are available, additional sensitivity analyses will be performed according to major model characteristics, such as graft type or reconstruction technique. These analyses aim to assess the consistency of treatment effects across different preclinical models and to inform the translational relevance of the findings.

Sensitivity analyses will be performed by sequentially omitting studies to evaluate the robustness of the results. Publication bias will be visually assessed using funnel plots when more than 10 studies are included, and further tested with Egger's regression test [12].

In case of missing data, we will follow the methodology outlined in the article [13] to address missing data in systematic reviews and meta-analyses. If data cannot be retrieved or imputed based on the methods outlined, we will report the missing data and conduct sensitivity analyses to evaluate the potential impact on the overall findings.

## 2.8. Ethics and dissemination

Ethical approval is not required because this study synthesizes published data only. Findings will be disseminated through peer-reviewed publication and conference presentations. Extracted data and analytic code will be made publicly available upon publication of the completed review.

## 3. Discussion

This systematic review and meta-analysis will provide a comprehensive synthesis of preclinical evidence regarding the role of stem cell-based therapies in tendon-bone healing following anterior cruciate ligament reconstruction. By integrating biomechanical, radiological, and histological outcomes across different animal models, this study will contribute to clarifying the potential mechanisms by which stem cells promote graft integration. The findings may offer valuable insights for translational research and provide a scientific basis for designing future clinical trials.

                                                    

The clinical relevance of this work is considerable. ACL rupture is one of the most common sports-related injuries, and failure of graft integration remains a major cause of poor outcomes and revision surgery. Stem cell-based strategies have been proposed as promising biological adjuvants to enhance tendon-bone healing, yet current evidence remains fragmented and inconsistent. By systematically pooling data from controlled animal studies, this review will help determine whether stem cells can consistently improve biomechanical strength and histological integration, thereby bridging the gap between preclinical research and clinical application.

The mechanisms through which stem cells promote tendon-bone healing after ACL reconstruction are multifaceted. First, stem cells, particularly mesenchymal stem cells (MSCs), are capable of secreting a variety of growth factors, which enhance osteogenesis and angiogenesis at the tendon-bone interface [14]. These factors stimulate the differentiation of surrounding cells into osteoblasts and endothelial cells, promoting bone formation and vascularization [15].

Furthermore, stem cells help to modulate the inflammatory response by secreting anti-inflammatory cytokines that reduce fibrosis and improve the tendon-bone interface integration [16]. By promoting tissue remodeling and reducing scar tissue formation, stem cells facilitate a smoother transition from tendon tissue to bone, improving the overall strength and stability of the graft. In addition, stem cell-derived products such as exosomes and extracellular vesicles have also been shown to carry bioactive molecules that mediate these beneficial effects, suggesting that the mechanisms extend beyond cell-based therapies alone.

Nevertheless, several limitations should be acknowledged. First, heterogeneity is expected due to variations in animal species, stem cell sources, delivery methods, and follow-up durations. Second, the methodological quality of animal studies may vary, and incomplete reporting of allocation concealment or blinding may introduce bias. Third, most included studies are short-term experiments, and long-term outcomes such as graft durability and joint function remain poorly evaluated. Finally, the external validity of animal models to human ACL reconstruction is inherently limited, and caution should be exercised when extrapolating the results to clinical practice.

Future studies should prioritise the use of standardised animal models, robust randomisation and blinding procedures, and clinically relevant endpoints. Moreover, head-to-head comparisons of different stem cell sources and delivery techniques are needed to identify the most effective strategies. In addition, further work should explore the underlying molecular mechanisms, such as paracrine signalling and immunomodulation, which may explain the beneficial effects of stem cells at the tendon-bone interface.

Overall, this review aims to summarise the current preclinical evidence, identify research gaps, and highlight the need for well-designed, standardised animal studies to facilitate the safe and effective translation of stem cell therapies into clinical ACL reconstruction.

## Supporting information

**S1 Table. PRISMA-P (Preferred Reporting Items for Systematic review and Meta-Analysis Protocols) 2015 checklist: recommended items to address in a systematic review protocol.**
(DOC)

**S2 Table. Search strategies for the systematic review and meta-analysis on stem cell therapies for tendon-bone healing after anterior cruciate ligament (ACL) reconstruction in animal models.**
(DOCX)

## Author contributions

**Conceptualization:** Shibo Zhao, Congcong Wang.

**Writing – original draft:** Shibo Zhao, Congcong Wang.

**Writing – review & editing:** Shibo Zhao, Congcong Wang.

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
