## [Decision Letter · Decision Letter 0]

7 Jan 2026

Dear Dr. Wang,

Thank you for submitting your manuscript to PLOS ONE. After careful consideration, we feel that it has merit but does not fully meet PLOS ONE’s publication criteria as it currently stands. Therefore, we invite you to submit a revised version of the manuscript that addresses the points raised during the review process.

We look forward to receiving your revised manuscript.

Kind regards,

Ramada Rateb Khasawneh

Academic Editor

PLOS One

Journal Requirements:

“Shandong Province medical health science and technology project (202404070285), Research on the Mechanism and Application of Microfragment Adipose tissue in promoting Tendon-bone Healing after anterior cruciate ligament Reconstruction”

Additional Editor Comments:

Conceptual ambiguity in intervention definition

Stem cell–based therapies, stem cell–derived products (e.g., exosomes), and scaffold-based delivery systems are grouped together without clearly stating that scaffolds are only included when combined with stem cells or their derivatives. This may introduce conceptual heterogeneity and should be explicitly clarified.

Potential redundancy and excessive length

The Introduction is relatively long for a protocol and includes detailed biological mechanisms and examples of individual animal studies that are not essential at this stage. Condensation would improve focus and readability.

Limited clarification on handling heterogeneity

Although subgroup analyses are planned, the protocol does not clearly specify how multiple outcome time points within the same study will be handled, nor how extreme heterogeneity will affect decisions about quantitative synthesis versus narrative reporting.

Incomplete specification of search strategy details

The protocol does not explicitly state whether language restrictions will be applied, nor does it clarify the systematic use of controlled vocabulary (e.g., MeSH/Emtree terms). Additionally, omission of databases such as Web of Science may limit search completeness.

Outcome hierarchy not fully defined

While outcomes are listed, the distinction between primary and secondary outcomes is not always explicit, particularly when both biomechanical and histological outcomes are pooled using standardized mean differences.

Limited methodological detail on data synthesis choices

The rationale for using standardized mean differences rather than mean differences is not explicitly justified, and handling of different histological scoring systems across studies is not fully explained.

Risk of bias assessment reporting

Although SYRCLE’s tool is appropriately selected, the protocol does not specify whether inter-reviewer agreement (e.g., kappa statistics) will be calculated, which may raise concerns about assessment reproducibility.

Translational limitations insufficiently operationalized

While the discussion acknowledges limited external validity of animal models, the protocol does not predefine how this limitation will be addressed analytically (e.g., species-based sensitivity analyses beyond basic subgrouping).

Inconsistencies and minor formatting issues

There are small inconsistencies in timelines (search end date vs study completion), occasional typographical errors, and spacing issues that may detract from overall polish.

Potential risk of selective reporting across outcomes

Given the wide range of possible biomechanical, imaging, and histological endpoints, clearer prespecification of which metrics will be prioritized for meta-analysis would reduce the risk of selective outcome reporting.

Reviewers' comments:

Reviewer's Responses to Questions

**Comments to the Author**

1. Does the manuscript provide a valid rationale for the proposed study, with clearly identified and justified research questions?

Reviewer #1: Yes

Reviewer #2: Yes

Reviewer #3: Yes

2. Is the protocol technically sound and planned in a manner that will lead to a meaningful outcome and allow testing the stated hypotheses?

Reviewer #1: Partly

Reviewer #2: Yes

Reviewer #3: Yes

3. Is the methodology feasible and described in sufficient detail to allow the work to be replicable?

Reviewer #1: Yes

Reviewer #2: Yes

Reviewer #3: Yes

4. Have the authors described where all data underlying the findings will be made available when the study is complete?

Reviewer #1: No

Reviewer #2: Yes

Reviewer #3: Yes

5. Is the manuscript presented in an intelligible fashion and written in standard English?

*PLOS ONE*

Reviewer #1: Yes

Reviewer #2: Yes

Reviewer #3: Yes

You may also provide optional suggestions and comments to authors that they might find helpful in planning their study.

Reviewer #1: 1) The authors must explicitly state a priori how these different comparisons will be handled. The protocol must specify if they will be analyzed in separate meta-analyses or, if they plan to combine them, provide a strong justification for doing so.

2) The subgroup analysis plan must be expanded to match the inclusion criteria. The authors must add a priori plans to analyze other cell sources (Synovial, Umbilical Cord) and, crucially, a separate subgroup for "stem cell-derived products (exosomes, EVs)" vs. "whole cell therapies."

3) The authors must add "cell dose" (e.g., total number of cells administered) to their data extraction plan. Furthermore, they must add a plan to conduct a meta-regression analysis to explore the dose-response relationship, as this is a highly likely and important source of heterogeneity.

4) Subgroup analyses for major, clinically obvious sources of heterogeneity (such as Control Type, Cell Type, and Dose, as mentioned above) should be planned a priori and performed regardless of the I² value. The I² > 75% threshold for post-hoc exploration is acceptable, but not for the pre-planned analyses.

5) Non-Compliant Data Availability Statement: The Data Availability Statement provided in the submission form ("The data underlying the results presented in the study are available from PubMed, Embase, Scopus, SPORTDiscus" ) is incorrect and misunderstands the PLOS policy.

This statement must be revised. The authors must declare where the data generated by this systematic review (i.e., the final extracted data spreadsheet used for the meta-analysis) will be made publicly available upon the study's completion (e.g., as a Supporting Information file or in a public data repository).

6) The "Keywords" list is improperly punctuated and formatted.

There is inconsistency in terminology: "Tendon-bone healing" (with a hyphen) and "Tendon bone healing" (without a hyphen) are used interchangeably, including in the search strategy itself. This must be standardized

7) If there is missing data, you can use this article https://doi.org/10.3390/app12031593.

Reviewer #2: The protocol is clearly written and addresses an important preclinical question. The objectives, eligibility criteria, and planned analyses are generally appropriate. I have several suggestions that would improve reproducibility, reduce the risk of selective synthesis, and align the protocol more closely with best practice for systematic review protocols.

1.Search end date: Please avoid the fixed “to August 2025” wording given screening starts in September 2025. Recommend “from inception to the date of the final search,” and report the exact final search date in the completed review.

2.Databases: Consider adding Web of Science (or justify exclusion) to improve capture of preclinical animal studies.

3.Multiple time points and outcomes: Please pre-specify how you will handle studies reporting multiple follow-up time points (e.g., separate analyses by prespecified time windows vs selecting the longest follow-up), and how multiple outcomes will be included/synthesized to minimize selective inclusion.

4.Effect size choice: Using SMD for all outcomes may reduce interpretability. Please specify MD for outcomes with consistent units (e.g., ultimate failure load, stiffness) and SMD (Hedges g) for outcomes measured on different scales (e.g., histology scores).

5.Follow-up subgrouping: The ≤4 vs >4 weeks threshold needs justification. Alternatively, consider more conventional strata (e.g., 4/8/≥12 weeks) aligned with common animal ACLR time points.

6.Data availability: Add a brief plan stating what will be shared (extracted dataset, risk-of-bias table, analysis code), where (e.g., OSF/Zenodo), and when (upon publication).

7.Minor clarity: Standardize terminology (tendon–bone vs tendon-to-bone healing), state language restrictions (or none), and for “full text cannot be retrieved,” note attempts via author contact/interlibrary loan before exclusion.

Reviewer #3: This article mentions that stem cells can promote tendon bone healing after cruciate ligament reconstruction, but the mechanism is not elucidated

**Do you want your identity to be public for this peer review?** For information about this choice, including consent withdrawal, please see our Privacy Policy

Reviewer #1: **Yes:** Esedullah AKARAS

Reviewer #2: No

Reviewer #3: No

---

## [Author Response · Author response to Decision Letter 1]

9 Jan 2026

Additional Editor Comments:

Conceptual ambiguity in intervention definition

Stem cell–based therapies, stem cell–derived products (e.g., exosomes), and scaffold-based delivery systems are grouped together without clearly stating that scaffolds are only included when combined with stem cells or their derivatives. This may introduce conceptual heterogeneity and should be explicitly clarified.

Thank you for your valuable feedback. We have clarified the conceptual ambiguity regarding the inclusion of scaffold-based delivery systems. In the revised manuscript, we explicitly state that scaffold-based delivery systems are only included when used in combination with stem cells or their derivatives, such as exosomes or extracellular vesicles. This clarification ensures that the intervention definition is consistent and avoids conceptual heterogeneity. We hope this resolves the issue and improves the clarity of the study design.

Potential redundancy and excessive length

The Introduction is relatively long for a protocol and includes detailed biological mechanisms and examples of individual animal studies that are not essential at this stage. Condensation would improve focus and readability.

Thank you for your valuable feedback. In response, we have condensed the introduction by removing detailed biological mechanisms and specific examples of individual animal studies that are not essential at this stage. We have focused more on the background of ACL injury, the potential of stem cell-based therapies, and the aim of our systematic review. We believe these revisions improve the focus and readability of the manuscript.

Limited clarification on handling heterogeneity

Although subgroup analyses are planned, the protocol does not clearly specify how multiple outcome time points within the same study will be handled, nor how extreme heterogeneity will affect decisions about quantitative synthesis versus narrative reporting.

Thank you for your feedback. In response, we have clarified how multiple outcome time points within the same study will be handled by prioritizing the longest follow-up period or conducting separate analyses for each relevant time point. Additionally, we have outlined how extreme heterogeneity will be addressed. If I² exceeds 75%, we will explore sources of heterogeneity through subgroup and sensitivity analyses. If substantial heterogeneity persists, we will report the results narratively rather than quantitatively. We hope these clarifications improve the methodology section of the protocol.

Incomplete specification of search strategy details

The protocol does not explicitly state whether language restrictions will be applied, nor does it clarify the systematic use of controlled vocabulary (e.g., MeSH/Emtree terms). Additionally, omission of databases such as Web of Science may limit search completeness.

Thank you for your feedback. In response, we have clarified that only English-language articles will be included in the search to ensure that only relevant English studies are considered. We have also added Web of Science to our search strategy and will systematically use controlled vocabulary (e.g., MeSH/Emtree terms) to ensure comprehensive coverage of the literature. We believe these modifications address your concerns and improve the search strategy.

Outcome hierarchy not fully defined

While outcomes are listed, the distinction between primary and secondary outcomes is not always explicit, particularly when both biomechanical and histological outcomes are pooled using standardized mean differences.

Thank you for this important comment. We have clarified the outcome hierarchy throughout the protocol. We now explicitly define biomechanical properties (ultimate failure load and stiffness) as the primary outcomes. Micro-CT–based measures of bone integration and histological outcomes are prespecified as secondary outcomes. In addition, we state that quantitative synthesis will prioritize primary outcomes, while secondary outcomes will be meta-analyzed only when sufficiently comparable data are available. These revisions improve transparency and reduce the risk of selective outcome reporting.

Limited methodological detail on data synthesis choices

The rationale for using standardized mean differences rather than mean differences is not explicitly justified, and handling of different histological scoring systems across studies is not fully explained.

Thank you for this important methodological comment. We have revised the data synthesis section to explicitly justify the choice between standardized mean differences (SMDs) and mean differences (MDs), and to clarify the handling of different histological scoring systems across studies.We now state that MDs will be used for outcomes reported with consistent units across studies, such as biomechanical measures (ultimate failure load and stiffness), to improve interpretability. SMDs will be applied when outcomes are measured using different scales or scoring systems.For histological outcomes, we clarify that quantitative synthesis will be conducted only when studies assess comparable constructs of tendon–bone interface healing. When different semi-quantitative scoring systems are used to evaluate similar constructs, SMDs (Hedges’ g) will be used to enable pooling. If histological scoring systems or outcome definitions are judged too heterogeneous for meaningful quantitative synthesis, results will be summarized narratively.

Risk of bias assessment reporting

Although SYRCLE’s tool is appropriately selected, the protocol does not specify whether inter-reviewer agreement (e.g., kappa statistics) will be calculated, which may raise concerns about assessment reproducibility.

Thank you for this helpful comment. We have revised the risk of bias assessment section to improve transparency and reproducibility. Specifically, we now state that inter-reviewer agreement for the SYRCLE risk of bias assessment will be evaluated using Cohen’s kappa statistic prior to consensus discussions, and the level of agreement will be reported. This addition clarifies the reliability of the assessment process.

Translational limitations insufficiently operationalized

While the discussion acknowledges limited external validity of animal models, the protocol does not predefine how this limitation will be addressed analytically (e.g., species-based sensitivity analyses beyond basic subgrouping).

Thank you for this insightful comment. We have revised the statistical analysis section to predefine how translational limitations of animal models will be addressed analytically. Specifically, we now state that prespecified sensitivity analyses will be conducted stratified by animal species, including repeating analyses after excluding individual species (e.g., rat-only or rabbit-only analyses) to assess the robustness of the pooled estimates. Where sufficient data are available, additional sensitivity analyses based on key experimental model characteristics will also be performed. These additions operationalize the assessment of external validity and improve the translational interpretability of the findings.

Inconsistencies and minor formatting issues

There are small inconsistencies in timelines (search end date vs study completion), occasional typographical errors, and spacing issues that may detract from overall polish.

Thank you for noting these issues. We have carefully reviewed the manuscript to improve overall consistency and presentation. We have corrected timeline inconsistencies by aligning the search end date and study timeline (i.e., revising the search period to “from inception to the date of the final search” and clarifying that the exact final search date will be reported in the completed review). In addition, we have corrected typographical errors, standardized terminology and formatting, and addressed spacing and punctuation issues throughout the manuscript to improve readability and polish.

Potential risk of selective reporting across outcomes

Given the wide range of possible biomechanical, imaging, and histological endpoints, clearer prespecification of which metrics will be prioritized for meta-analysis would reduce the risk of selective outcome reporting.

Thank you for this important comment. We have revised the data synthesis section to more clearly prespecify which outcome metrics will be prioritized for meta-analysis in order to reduce the risk of selective outcome reporting.We now explicitly state that outcomes will be synthesized according to a prespecified hierarchy. Biomechanical outcomes (ultimate failure load and stiffness) are defined as primary outcomes and will be prioritized for meta-analysis. Secondary outcomes, including micro-CT measures and histological outcomes, will be synthesized only when at least two studies report sufficiently comparable data.To further minimize selective inclusion, we prespecify the prioritization of outcome metrics within each outcome domain. For histological outcomes, only measures assessing comparable constructs of tendon–bone interface healing will be pooled quantitatively. When multiple histological domains are reported within the same study, we will extract the prespecified overall score where available; otherwise, the domain most consistently reported across studies will be prioritized according to predefined rules. Outcomes judged to be too heterogeneous for meaningful quantitative synthesis will be summarized narratively.These prespecified analytic rules improve transparency and substantially reduce the risk of selective outcome reporting.

Reviewer #1: 1) The authors must explicitly state a priori how these different comparisons will be handled. The protocol must specify if they will be analyzed in separate meta-analyses or, if they plan to combine them, provide a strong justification for doing so.

Thank you for this comment. We have revised the protocol to state a priori how heterogeneous comparisons will be handled. We will conduct meta-analyses separately by intervention category, primarily distinguishing live stem cells from stem cell–derived products (e.g., exosomes/extracellular vesicles). We will present an overall pooled estimate across stem cell–based therapies only when studies are sufficiently comparable regarding intervention category, control condition, outcome definition, and follow-up timing; otherwise, results will be synthesized narratively.

We also added an explicit plan for multi-arm studies to avoid double-counting shared control groups (combining intervention arms when appropriate, or splitting the shared control group across comparisons when arms must be retained separately). These changes were added to the Methods (Section 2.7).

2) The subgroup analysis plan must be expanded to match the inclusion criteria. The authors must add a priori plans to analyze other cell sources (Synovial, Umbilical Cord) and, crucially, a separate subgroup for "stem cell-derived products (exosomes, EVs)" vs. "whole cell therapies."

Thank you for this suggestion. We have expanded our a priori subgroup analysis plan to align with the eligibility criteria. Specifically, we now prespecify subgroup analyses by therapy format (i.e., whole-cell therapies versus stem cell–derived products such as exosomes/extracellular vesicles) and by cell source for whole-cell therapies, explicitly including synovial-derived and umbilical cord/umbilical cord blood–derived cells in addition to bone marrow–, adipose-, and tendon-derived sources. These revisions have been incorporated in the Methods (Section 2.7, Subgroup analyses).

3) The authors must add "cell dose" (e.g., total number of cells administered) to their data extraction plan. Furthermore, they must add a plan to conduct a meta-regression analysis to explore the dose-response relationship, as this is a highly likely and important source of heterogeneity.

Thank you for this valuable suggestion. We agree that administered dose is a plausible and important source of heterogeneity in preclinical stem cell studies. We have therefore updated the protocol to (i) explicitly extract dose information (including the total number of cells administered, and where necessary concentration/volume and number of administrations to derive total dose) in the Data extraction section, and (ii) prespecify a random-effects meta-regression to explore the dose–response relationship when sufficient studies are available (typically ≥10). Meta-regression will be restricted to studies reporting or allowing derivation of a comparable total cell dose; otherwise, dose effects will be summarized narratively. These changes were added to Methods (Sections 2.5 and 2.7).

4) Subgroup analyses for major, clinically obvious sources of heterogeneity (such as Control Type, Cell Type, and Dose, as mentioned above) should be planned a priori and performed regardless of the I² value. The I² > 75% threshold for post-hoc exploration is acceptable, but not for the pre-planned analyses.

Thank you for your valuable suggestion. We agree that major sources of heterogeneity, such as control type, cell type, and dose, should be considered a priori in subgroup analyses, regardless of the I² value. We have revised the protocol accordingly to explicitly state that prespecified subgroup analyses will be conducted for these clinically obvious sources of heterogeneity, and these analyses will be performed regardless of the I² value, provided that sufficient data are available.

Additionally, we retained the I² > 75% threshold for exploratory analyses after conducting the prespecified subgroup analyses, and will explore further sources of heterogeneity through meta-regression or narrative synthesis if appropriate. These changes have been incorporated into the Statistical analysis and data synthesis section (Methods, Section 2.7).

5) Non-Compliant Data Availability Statement: The Data Availability Statement provided in the submission form ("The data underlying the results presented in the study are available from PubMed, Embase, Scopus, SPORTDiscus" ) is incorrect and misunderstands the PLOS policy.

This statement must be revised. The authors must declare where the data generated by this systematic review (i.e., the final extracted data spreadsheet used for the meta-analysis) will be made publicly available upon the study's completion (e.g., as a Supporting Information file or in a public data repository).

Thank you for pointing out the issue with the Data Availability Statement. We have revised the statement to clarify that the data underlying the results (the final extracted data used for the meta-analysis) will be made publicly available upon publication. The data will be provided either as Supporting Information files or stored in a public data repository (e.g., Open Science Framework, Figshare, Dryad), in accordance with PLOS policy. This update has been made in the Data Availability Statement section of the manuscript.

6) The "Keywords" list is improperly punctuated and formatted.

There is inconsistency in terminology: "Tendon-bone healing" (with a hyphen) and "Tendon bone healing" (without a hyphen) are used interchangeably, including in the search strategy itself. This must be standardized

Thank you for pointing out the inconsistency in the punctuation and terminology. We have standardized the use of "Tendon-bone healing" throughout the manuscript, including in the Keywords and Search strategy sections. The punctuation and formatting issues have been corrected to ensure consistency.

7) If there is missing data, you can use this article https://doi.org/10.3390/app12031593.

Thank you for the suggestion. We have added a statement in the Data Analysis section specifying that if missing data are encountered, we will refer to the methodology outlined in the article https://doi.org/10.3390/app12031593 to handle the issue. If imputation is not possible, we will conduct sensitivity analyses to assess the potential impact of missing data on our findings.

Reviewer #2: The protocol is clearly written and addresses an important preclinical question. The objectives, eligibility criteria, and planned analyses are generally appropriate. I have several suggestions that would improve reproducibility, reduce the risk of selective synthesis, and align the protocol more clo

---

## [Decision Letter · Decision Letter 1]

13 Jan 2026

Effect of stem cell therapies on tendon–bone healing after anterior cruciate ligament reconstruction in animal models: protocol for a systematic review and meta-analysis

PONE-D-25-50568R1

Dear Dr. Wang,

We’re pleased to inform you that your manuscript has been judged scientifically suitable for publication and will be formally accepted for publication once it meets all outstanding technical requirements.

Kind regards,

Ramada Rateb Khasawneh

Academic Editor

PLOS One

Additional Editor Comments (optional):

Good Luck

Reviewers' comments:

Reviewer's Responses to Questions

**Comments to the Author**

1. Does the manuscript provide a valid rationale for the proposed study, with clearly identified and justified research questions?

Reviewer #1: Yes

Reviewer #2: Yes

2. Is the protocol technically sound and planned in a manner that will lead to a meaningful outcome and allow testing the stated hypotheses?

Reviewer #1: Yes

Reviewer #2: Yes

3. Is the methodology feasible and described in sufficient detail to allow the work to be replicable?

Reviewer #1: Yes

Reviewer #2: Yes

4. Have the authors described where all data underlying the findings will be made available when the study is complete?

Reviewer #1: Yes

Reviewer #2: Yes

5. Is the manuscript presented in an intelligible fashion and written in standard English?

*PLOS ONE*

Reviewer #1: Yes

Reviewer #2: Yes

You may also provide optional suggestions and comments to authors that they might find helpful in planning their study.

Reviewer #1: The article is now ready for acceptance after the necessary corrections have been made.

Reviewer #2: The author has written an excellent article. After improving the relevant details, I believe it meets the publication requirements of PLOS ONE.

**Do you want your identity to be public for this peer review?** For information about this choice, including consent withdrawal, please see our Privacy Policy

Reviewer #1: No

Reviewer #2: No

---

## [Editor Report · Acceptance letter]

PONE-D-25-50568R1

PLOS One

Dear Dr. Wang,

I'm pleased to inform you that your manuscript has been deemed suitable for publication in PLOS One. Congratulations! Your manuscript is now being handed over to our production team.

Kind regards,

on behalf of

Dr. Ramada Rateb Khasawneh

Academic Editor

PLOS One